# Enhanced NaFe_0.5_Mn_0.5_O_2_/C Nanocomposite as a Cathode for Sodium-Ion Batteries

**DOI:** 10.3390/nano12060984

**Published:** 2022-03-16

**Authors:** Murugan Nanthagopal, Chang Won Ho, Nitheesha Shaji, Gyu Sang Sim, Murugesan Varun Karthik, Hong Ki Kim, Chang Woo Lee

**Affiliations:** 1Department of Chemical Engineering (Integrated Engineering), College of Engineering, Kyung Hee University, 1732 Deogyeong-daero, Giheung, Yongin 17104, Gyeonggi, Korea; nanthamurugan@khu.ac.kr (M.N.); ghckddnjs@khu.ac.kr (C.W.H.); nitheesha@khu.ac.kr (N.S.); simgyusang0215@khu.ac.kr (G.S.S.); varun219@khu.ac.kr (M.V.K.); hkkim95@khu.ac.kr (H.K.K.); 2Center for the SMART Energy Platform, College of Engineering, Kyung Hee University, 1732 Deogyeong-daero, Giheung, Yongin 17104, Gyeonggi, Korea

**Keywords:** sodium-ion battery, energy materials, combustion, energy storage, layered-type material

## Abstract

Sodium-ion batteries (SIBs) have emerged as an alternative candidate in the field of energy storage applications. To achieve the commercial success of SIBs, the designing of active materials is highly important. O3-type layered-NaFe_0.5_Mn_0.5_O_2_ (NFM) materials provide higher specific capacity along with Earth-abundance and low cost. Nevertheless, the material possesses some disadvantages, such as a low rate capability and severe capacity fading during cycling. To overcome such drawbacks, composite O3-type layered NFM with carbon has been prepared for the cathode electrode of SIBs through a facile solution combustion method followed by calcination process. The introduction of carbon sources into NFM material provides excellent electrochemical performances; moreover, the practical limitations of NFM material such as low electrical conductivity, structural degradation, and cycle life are effectively controlled by introducing carbon sources into the host material. The NFM/C-2 material delivers the specific charge capacities of 171, 178, and 166 mA h g^−1^; and specific discharge capacities of 188, 169, and 162 mA h g^−1^, in the first 3 cycles, respectively.

## 1. Introduction

Rechargeable batteries are among the most successful systems for storing electricity and supplying power to energy devices in existing energy storage technologies. Lithium-ion batteries (LIBs) are commercially used for their energy density, excellent conversion performance, and easy maintenance, and are expected to be a suitable option for vehicles [1,2,3,4,5]; nevertheless, the scarcity and cost of lithium sources have created the need to find alternatives to LIBs [6,7,8]. A growing interest in sodium-ion batteries (SIBs) for energy storage applications can be attributed to sodium’s high energy density, abundance on Earth, and cost-effectiveness. Moreover, the rocking chair mechanism of sodium storage is quite similar to that of LIBs [9,10,11]. In particular, it is believed that SIBs have some of the most favorable contenders for large-grid applications. In addition, SIBs do not require a copper current collector for the anode materials, which means that the weight and cost of the battery can be reduced significantly [12,13,14,15]. Researchers in the energy storage community believe that SIBs can compete with LIBs in responding to future energy demands despite their similar mechanisms. SIBs, on the other hand, have a lower energy density than LIBs., which makes them less suitable for automobile and portable applications [16,17,18,19]. This is because of their larger ionic radius (1.02 Å for Na^+^ vs. 0.67 Å for Li^+^), higher atomic weight (23 for Na vs. 7 for Li), and lower standard potential (2.71 V for Na vs. 3.04 V for Li) [20,21,22]. To date, various components, such as phosphates, fluorophosphates, and layered transition metal oxides have been tested as cathodes for sodium-ion storage devices. Among these components, the SIB cathode materials that use layered transition metal oxides have been identified as potential candidates. [23,24,25,26,27].

Layered-type sodium transition metal oxides (NaTMO_2_, where TM = Mn, Fe, Ni, V, Ti, etc.) can provide higher capacities, faster Na^+^-ion diffusion, and high energy densities. In accordance with the notation from Delmas, NaTMO_2_ is categorized into two types, O3 and P2, according to different oxygen stacking orderings, where Na^+^ is located in a unit cell between layers in the octahedral (O) and prismatic (P) sites [28,29,30,31]. Compared with P2-type transition metal oxides, O3-type materials are promising candidates as a cathode for an SIB full cell. In particular, O3-type layered cathode materials facilitate Na^+^ insertion into the material, enabling the fabrication of practical sodium-ion full cells with hard carbon as an anode material. Different cathodes of O3-type layered materials have been intensively studied. The oxygen stacking is different in the O3, P2, and P3-type structures. For instance, the oxygen stacking of O3, P2, and P3-type structures is ABCABC, ABBA, and ABBCCA, respectively [32,33,34]; however, because of the Na^+^ ion greater radius, many structural transformations are inevitable in the host structure during charging/discharging, leading to poor cyclic performance and low energy efficiency [35,36,37]. Layered NaMnO_2_ provide significant energy density, and Fe is electrochemically active in the Na-ion cathode. Researchers have attempted Fe doping in NaMnO_2_ to improve the electrochemical performance through a synergistic effect [38,39]. Partial Fe doping in NaMnO_2_ will provide higher initial capacities; however, the material has limitations, such as low rate performance, and huge capacity fading during cycling. To address these issues, several scientific techniques have been used on the material. [40,41,42,43,44].

In this study, we developed an O3-type layered NaFe_0.5_Mn_0.5_O_2_ (NFM) nanocomposite material through a facile solution combustion technique followed by calcination. The as-synthesized material was applied as a cathode for the SIB half-cell. The material exhibits higher specific charge/discharge capacity in initial cycles at a lower current rate; however, it suffers from poor rate capability and drastic capacity fading when cycled at higher current rates, which limits its potential applications. To overcome these limitations, we applied carbon into the host NFM material to form a composite, which was then used as the cathode in the sodium-ion energy storage device. To the best of our knowledge, this is the first time that the physical, chemical and electrochemical properties of as-synthesized NFM/carbon composite material have been investigated and reported. Preparation of NFM active material using solution combustion method will also be newly discussed.

## 2. Materials and Methods

### 2.1. Chemicals

The chemicals used are as follows: sodium nitrate (NaNO_3_) and 2-methylimidazole (CH_3_C_3_H_2_N_2_H) from Sigma Aldrich, Seoul, Korea; manganese (II) nitrate hexahydrate (Mn(NO_3_)_2_·6H_2_O), and iron (III) nitrate enneahydrate (Fe(NO_3_)_3_·9H_2_O) from Junsei Chemical, Tokyo, Japan; and glycine fuel (C_2_H_5_NO_2_) from Dae-Jung Chemicals, Siheung-si, Korea. All reagents were analytical and used directly in the reactions without any further treatment.

### 2.2. Preparation of Sodium Iron Manganese Oxide/Carbon Composite NaFe_0.5_Mn_0.5_O_2_/C

The NaFe_0.5_Mn_0.5_O_2_/carbon composite materials were prepared by a facile solution combustion synthesis technique, continued via dry solid-state technique. Figure 1 represents the schematic illustration of NaFe_0.5_Mn_0.5_O_2_/carbon nanocomposite materials.

#### 2.2.1. Step 1: Synthesis of Sodium Iron Manganese Oxide

The solution combustion synthesis of NaFe_0.5_Mn_0.5_O_2_ material is shown in Figure 2a–c. Stoichiometric amounts (1:0.5:0.5) of sodium, iron, manganese precursors were completely dissolved in 5 mL of deionized water. C_2_H_5_NO_2_ as a fuel for the combustion reaction was then added to the solution, followed by a 5% sodium precursor, which was added to manage the thermal loss of sodium during calcination. Precursor compounds were entirely dissolved, and the resulting solution was heated on a heating plate at 250 °C. The nitrous gas was released, and complete combustion was achieved after the solution reaches its ignition point. The material was collected and heated at 450 °C for 3 h in an air atmosphere to break the nitrate and organic compounds from the precursor. After the temperature of the material decreased to room temperature, the precursors were finely ground and then calcined again at 950 °C for 15 h in an air atmosphere. The NFM nanocomposite material formed was transferred to an inert atmosphere to prevent any reactions with the ambient atmosphere.

#### 2.2.2. Step 2: Preparation of NaFe_0.5_Mn_0.5_O_2_/C Composite

The NFM/carbon composite was prepared by a facile single-step solid-state method, followed by pyrolysis. Typically, 2 g of as-synthesized NFM was added to a specific amount of 2-methylamidazole and ground for approximately 45 min to achieve proper mixing. Then, the NFM/C-containing precursors were calcined at 550 °C for 4 h in argon (Ar) atmosphere. For comparative studies, different weight percentages (1, 3, and 5 wt.%) of carbon precursors were used. The samples with 1, 3, and 5 wt.% carbon materials are called NFM/C-1, NFM/C-2, and NFM/C-3, respectively. The schematic illustration of the NFM and NFM/C composite electrode surfaces are shown in Figure 2d.

### 2.3. Physicochemical Characterization

Using an X-ray diffractometer (XRD, D8 advance, Bruker, Billerica, MA, USA), the phase purity and crystallinity of the as-synthesized bare NFM and NFM/C composite materials were investigated. A field-emission scanning electron microscope (FE-SEM; LEO SUPRA 55, GENESIS 2000 Carl Zeiss, Jena, Germany, EDAX, Mahwah, NJ, USA) and field-emission transmission electron microscope (FE-TEM, JEM-2100F, JEOL, Tokyo, Japan) fitted with an energy dispersive spectrometer (EDS; dual silicon drift detector(SDD)-type) were used to perform the elemental mapping of the prepared samples. X-ray photoelectron spectroscopy (XPS; K-Alpha, Thermo Fisher, Waltham, MA, USA) was also used to determine the elemental compositions and electronic valence states.

### 2.4. Electrochemical Characterization

The NFM or NFM/C as a working electrode for SIBs was prepared using N-methyl-2-pyrrolidone (NMP) as a solvent to create a slurry that combines 80% as-synthesized material as the active material, 10% polyvinyl difluoride, and 10% denka black. The slurry was coated using the doctor blade method on battery-grade aluminum foil and allowed to dry naturally. The dry electrode was then heated at 120 °C for 5 h in an oven to remove the NMP content in the electrode. The heated electrode was roll-pressed and punched into 14 mm circular disks to serve as the positive electrode and finally vacuum dried at 100 °C for 5 h. Electrochemical investigations of the as-synthesized cathode electrode were conducted using CR2032-type coin cells assembled inside an Ar-filled glove box. The as-synthesized NFM or NFM/C composite electrode was used as the cathode. A counter electrode made from pure sodium metal foil was used, the separator was used with a Whatman glass microfiber, and a mixture of ethylene carbonate and diethylene carbonate (1:1 volume ratio) with 5 wt.% fluoroethylene carbonate was dissolved in a 1 M NaClO_4_ electrolyte. An electrochemical cycler (Battery Tester 05001, HTC, Mumbai, India) was used to analyze the galvanostatic discharge/charge properties in the voltage range of 1.5–4.3 V. Differential capacity vs. voltage (dQ/dV) curves were studied by using the Iviumstat (Ivium technologies, Eindhoven, The Netherlands) electrochemical workstation.

## 3. Results and Discussion

XRD analysis was performed to identify the crystallinity and phase purity of the as-synthesized materials. Figure 3a shows the diffraction patterns, which are well matched with JCPDS No. 053-0349 with a rhombohedral crystal system that possesses an R-3m space group [45]. The sharp peaks indicate that the materials are highly crystalline in nature, and no impurities were observed in these samples. The as-prepared O3-NFM nanocomposite material attains an average crystalline size of about 42.69 nm. This high crystallinity helps the material exhibit a highly reversible structural behavior during Na^+^ intercalation and de-intercalation. The peaks at 16.05° (003) and 41.45° (104) classify them as an O3-type layered NFM material [46,47]. Notably, the peak at 44.71° (015) was missing for the carbon composite materials owing to the introduction of carbon to the surface of NFM material, while the peak at 16.05° (003) is slightly shifted to higher angles in the NFM/C composite materials. The XRD results confirm the formation of O3-NFM material with a fine crystal structure through a facile solution combustion technique. It is important to store the material in an inert atmosphere to prevent the oxidation reaction with H^+^/Na^+^ or water upon exposure to air. To determine the size and morphological structure, the as-synthesized NFM, NFM/C-1, NFM/C-2, and NFM/C-3 materials were analyzed by FE-SEM; the images obtained are shown in Figure 3. All the materials obtained irregular morphologies with a size between 3 and 5 µm. Figure 3b,c shows the FE-SEM image of NFM nanocomposite, which reveals the irregular shape of the material. The magnified image shows that the surface of the material is smooth with no observed agglomerations. Figure 3 shows the FE-SEM images of the NFM/C-1 (Figure 3d,e), NFM/C-2 (Figure 3f,g), and NFM/C-3 (Figure 3h,i) materials, respectively. In the NFM/C-2 and NFM/C-3 nanocomposite materials, the carbon particles are present in the surface cavities of the NFM material. Because the weight percentage of the carbon material is lower in the NFM/C-1 composite material, the surface of the material is similar to that of the bare NFM material. The solid-state method followed by calcination produces NFM/carbon nanocomposite.

To confirm the formation of the NFM/C composite, the NFM/C-2 material was characterized by FE-TEM equipped with dual SDD-type EDS mapping. Figure 4 presents the FE-TEM image with the elemental distribution and EDS mapping spectra of the NFM/C-2 material. Figure 4a shows the FE-TEM image of the NFM/C-2 material, while Figure 4b–f shows the elemental distributions of Na, Fe, Mn, O, and C elements, respectively. The elemental distributions indicate that all elements are evenly distributed in the NFM/C-2 material. Furthermore, elemental mapping confirmed the presence of all the elements in the compound, as shown in Figure 4g. The chemical and valence states of all the elements in the as-synthesized NFM/C-2 material was investigated by XPS characterization, and the results are shown in Figure 5. As expected, the survey spectrum of the as-synthesized NFM/C-2 material indicates the presence of Na, Fe, Mn, O, and C, as shown in Figure 5a. Figure 5b–f represents the core spectrum of Na, Fe, Mn, O, and C, respectively. Figure 5f represents the core spectra of C 1s, in which the peaks at the binding energies of 284.00 and 288.42 eV indicate that different chemical states of carbon are present in the as-synthesized material. The peak at the binding energy of 284.00 eV confirms that the material possesses C–C bond and the peak at 288.42 eV indicates the O–C=O bond. These two types of carbon bonding are beneficial for electrochemical reactions in batteries. The XPS results confirm that the solid-state method can effectively produce an NFM/C composite. These results strengthen the EDS mapping results.

Figure 6 shows the dQ/dV curves of the as-prepared NFM, NFM/C-1, NFM/C-2, and NFM/C-3 materials, respectively. The as-prepared materials have two redox peaks. The NFM material shows the oxidation peaks at 2.42 V and 3.92 V during the charge, and it shows 2 reduction peaks at around 3.50 V and 2.15 V during the discharge after the initial cycle, as shown in Figure 6a. As shown in Figure 6b, the NFM/C-1 material, the redox peaks are slightly moved to a higher voltage than the NFM material. Figure 6c,d represents the dQ/dV curves of the as-prepared NCM/C-2 and NFM/C-3, respectively. The redox peaks in terms of voltage for the carbon composite materials are different than NFM material, it is due to the lesser Na content on the composite materials that are hindered by the carbon particles on the surface of the composite.

Figure 7 shows the electrochemical performance of the as-synthesized materials as a cathode for SIBs. These were analyzed by fabricating a half-cell with pure sodium metal foil as the counter electrode, and the as-synthesized NFM as the working electrode. Galvanostatic charge/discharge studies were conducted on the materials with a cut-off voltage of 1.5–4.3 V at a current rate of 0.05 C. Figure 7a–d shows the galvanostatic charge/discharge profiles of first three cycles for the NFM, NFM/C-1, NFM/C-2, and NFM/C-3 materials, respectively. As seen in Figure 7a, the NFM material exhibits the initial specific charge/discharge capacity of 176/172 mA h g^−^^1^ with a coulombic efficiency of 97.8%. In the second cycle, the specific charge/discharge capacity dropped to 156/151 mA h g^−^^1^ with a coulombic efficiency of 96.4%. In the third cycle, the specific charge/discharge capacity faded to 148/144 mA h g^−^^1^ with a coulombic efficiency of 97.4%. At a low current rate, the NFM material delivered higher specific capacities for the initial cycle with higher coulombic efficiencies. As seen in Figure 7b, the NFM/C-1 material delivers the specific charge capacities of 139, 148, and 141 mA h g^−^^1^; and specific discharge capacities of 157, 141, and 139 m Ah g^−^^1^, in the first 3 cycles, respectively. As shown in Figure 7c, the NFM/C-2 material delivers the specific charge capacities of 171, 178, and 166 mA h g^−^^1^; and specific discharge capacities of 188, 169, and 162 mA h g^−^^1^, in the first 3 cycles, respectively. As seen in Figure 7d, the NFM/C-3 material delivers specific charge capacities of 153, 165, and 154 mA h g^−^^1^; specific discharge capacities of 169, 157, and 150 mA h g^−^^1^ in the first 3 cycles, respectively. Remarkably, the NFM/C-2, composite material exhibited higher specific capacities compared with NFM and other carbon composite materials. It should be noted that the initial charge profile of the NFM/C composite is different from that of NFM because of the stronger solid electrolyte interphase layer formed on the anode surface. The coulombic efficiency of the first cycle appears to be abnormal at over ~100% because of the lower initial sodium content, which was hindered by the carbon content in the composite material [48,49]. It is evident, therefore, that the formation of a composite with carbon can affect the electrochemical reaction and eventually lead to improving the performance of NFM cathodes for sodium-ion storage.

To understand the stability of the material, the cyclic performance was studied in the voltage range of 1.5–4.3 V at the current rate of 0.5 C with 2 formation cycles at 0.1 C. The cyclic performance of the materials is shown in Figure 7e. After the formation cycles, the bare NFM material delivered the specific discharge capacity of 110 mA h g^−^^1^ at 0.5 °C, however, the specific discharge capacity of the NFM material faded drastically to 24 mA h g^−^^1^ for 40 cycles. It was previously reported that the sodium layered-oxide cathodes often undergo side reactions with a liquid electrolyte and have poor structural stability above 4.0 V, which significantly reduces Na^+^ ion transport during cycling [50,51]. After the formation cycles, the specific discharge capacities of NFM/C-1, NFM/C-2, and NFM/C-3 were 114, 113, and 112 mA h g^−^^1^, respectively. At 100 cycles, the specific discharge capacities of NFM/C-1, NFM/C-2, and NFM/C-3 were 47, 75, and 68 mA h g^−^^1^, respectively. After 100 cycles at 0.5 C, the NFM/C-1, NFM/C-2, and NFM/C-3 materials had capacity retentions of 41.22, 66.79, and 60.59%, respectively. The NFM/C composite materials provide better cyclic performance than NFM material; it is evident that the carbon composites increased the structural stability and reduced the degradation of the host material. The rate capability was investigated at different current rates, as shown in Figure 7f. Different current rates (e.g., 0.05, 0.1, 0.5, 1, 2, and 5 C) were applied to the as-synthesized materials. For the NFM material, the capacity was close to zero at the current rate of 5 C; it regained its capacity to 104 mA h g^−^^1^ at 0.1 C, which faded to 74 mA h g^−^^1^ after a few cycles. The NFM/C composite materials exhibited better rate performance than the bare NFM. Among the composite materials, the NFM/C-2 material also provided better rate performance. Even at a higher current rate of 5 C, it delivered the capacity of 40 mA h g^−^^1^; importantly, it has good reversibility at different current rates.

## 4. Conclusions

We report O3-type layered NFM cathode materials that have been synthesized through a facile solution combustion technique for sodium-ion storage. The carbon with NFM material as a nanocomposite was prepared through a facile solid-state method to enhance the electrochemical properties. The physicochemical properties of as-synthesized NFM/C composite materials were analyzed through various methods. The NFM/C nanocomposite materials exhibited excellent electrochemical performance as a cathode for SIBs. Among the various weight percentages of the NFM/C nanocomposite materials, NFM/C-2 with 3 wt.% carbon showed the best electrochemical performance. After 100 cycles, the material delivered 75 mA h g^−1^ at the current rate of 0.5 C; even in terms of rate performance, the NFM/C-2 material delivered improved capacities at different current rates. Above all, the attempted NFM/C-2 has delivered a high level of electrochemical performances in terms of capacity and capacity retention, although it still showed limited values when compared to the sodium layered-oxide cathode materials reported so far; therefore, the carbon composite helps to improve the structural stability of NFM cathode materials at higher current rates due to the increase of the electrical conductivity along with the Na^+^ mobility for sodium-ion storage and the NFM/C nanocomposite is a promising candidate for high-performance SIBs.

## Figures and Tables

**Figure 1 nanomaterials-12-00984-f001:**
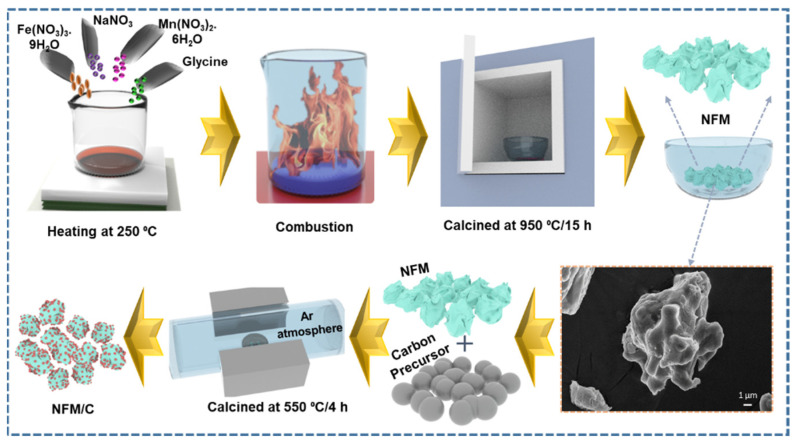
Schematic illustration of the preparation of NFM and NFM/C composite materials.

**Figure 2 nanomaterials-12-00984-f002:**
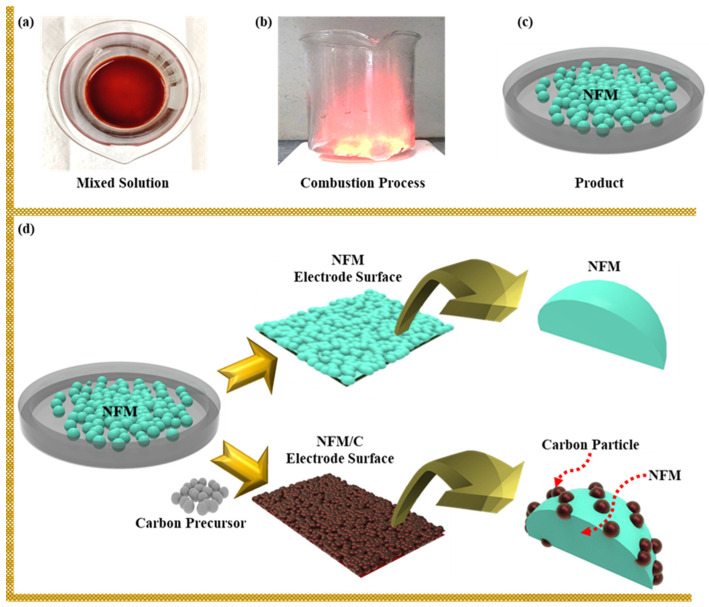
Schematic illustration: (**a**–**c**) solution combustion synthesis of NFM material, and (**d**) electrode surfaces of NFM and NFM/C-2.

**Figure 3 nanomaterials-12-00984-f003:**
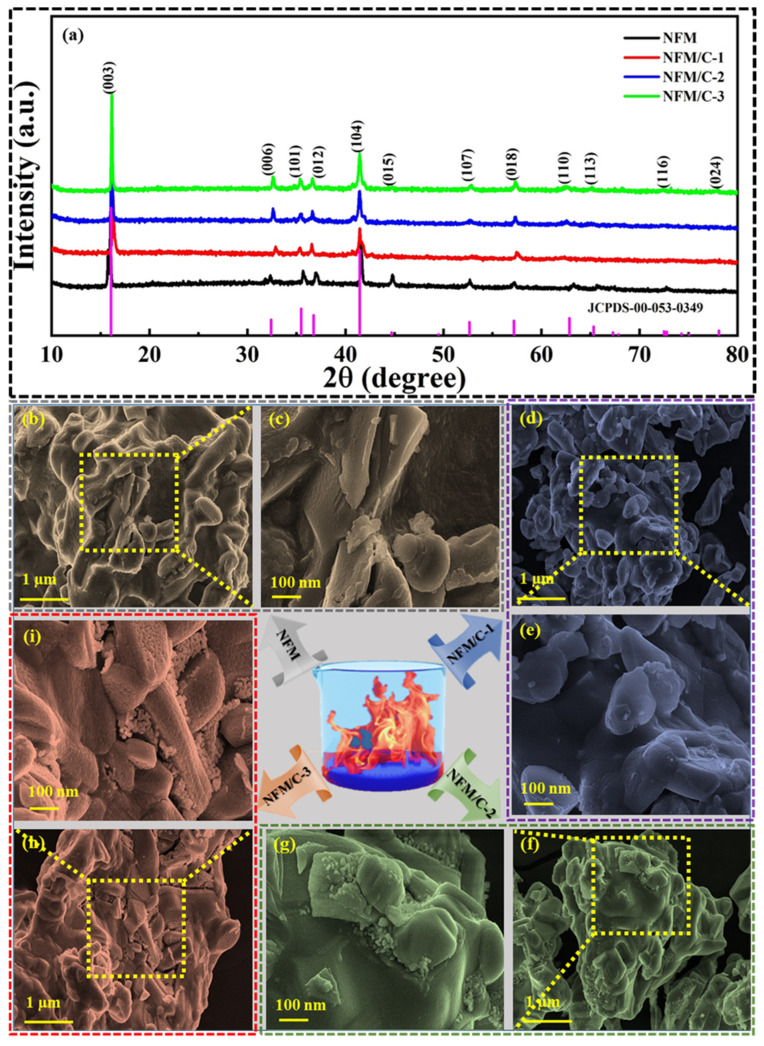
XRD patterns (**a**) for as-prepared NFM, NFM/C-1, NFM/C-2, and NFM/C-3 materials. FE-SEM images at various magnifications: (**b**,**c**) NFM, (**d**,**e**) NFM/C-1, (**f**,**g**) NFM/C-2, and (**h**,**i**) NFM/C-3 materials, respectively.

**Figure 4 nanomaterials-12-00984-f004:**
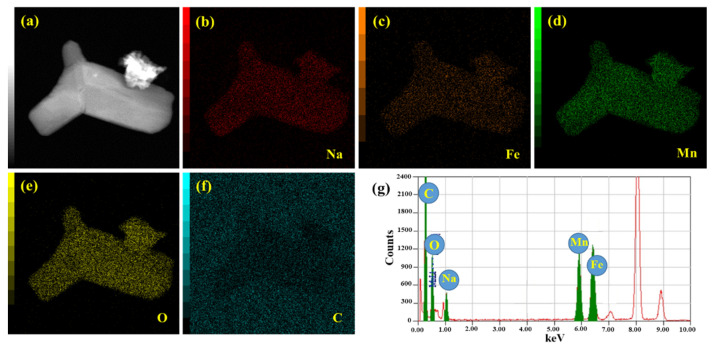
FE-TEM image (**a**) of as-prepared NFM/C-2 material, elemental distributions (**b**–**f**) of Na, Fe, Mn, O, and C, respectively, and EDS elemental spectrum (**g**) for as-prepared NFM/C-2 material.

**Figure 5 nanomaterials-12-00984-f005:**
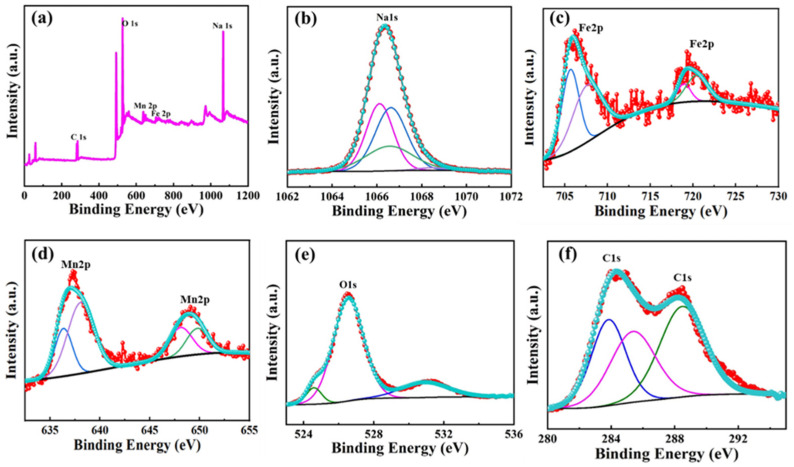
XPS spectra of as-prepared NFM/C-2: (**a**) survey spectrum, core spectrum of (**b**) Na 1s, (**c**) Fe 2p, (**d**) Mn 2p, (**e**) O 1s, and (**f**) C 1s.

**Figure 6 nanomaterials-12-00984-f006:**
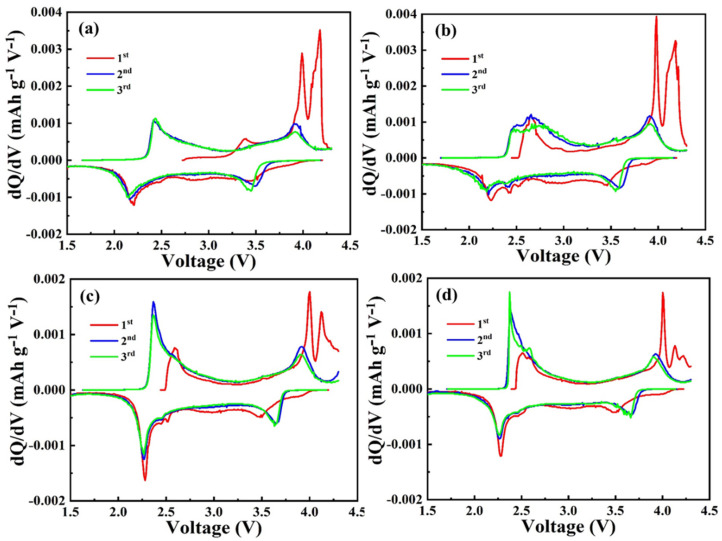
Differential capacity vs. voltage (dQ/dV) curves of as-prepared materials: (**a**) NFM, (**b**) NFM/C-1, (**c**) NFM/C-2, and (**d**) NFM/C-3.

**Figure 7 nanomaterials-12-00984-f007:**
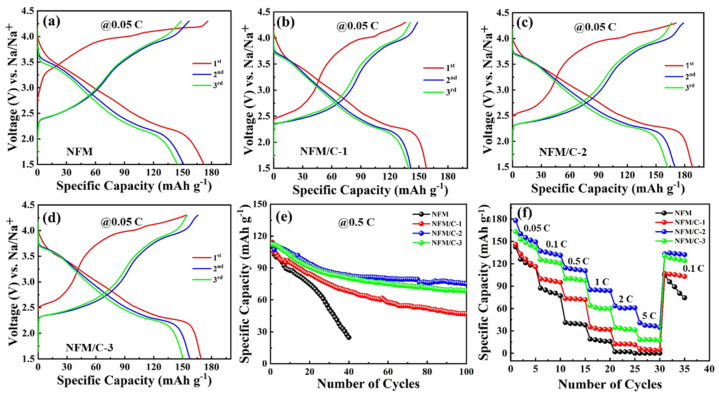
Electrochemical performances of as-prepared NFM, NFM/C-1, NFM/C-2, and NFM/C-3 materials, respectively: (**a**–**d**) galvanostatic charge/discharge curve at a current rate of 0.05 C for first 3 cycles, (**e**) cyclic performances at 0.5 C rate, and (**f**) rate capability at different current rates.

## Data Availability

Not applicable.

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
