# Peer review of "Enhanced NaFe0.5Mn0.5O2/C Nanocomposite as a Cathode for Sodium-Ion Batteries"

_nanomaterials, 2022, doi:10.3390/nano12060984_

Round 1

Reviewer 1 Report

In this paper, the authors prepared composite layered NFM with carbon  for the cathode electrode of SIBs through a facile solution combustion method followed by calcination process. This material present an excellent electrochemical performances. This work is written well and organized reasonably. I wonder to recommend it for publication in Nanomaterials after some revisions.

  1. This work provided the XPS data in fig. 5. It is better if the authors could add some discussion on the interface adhesion properties between C and NFM. Please see the references: Extreme Mechanics Letters 9 (2016) 226-236; Journal of Power Sources 290 (2015) 114-122.
  2. The carbon particles only attach to the NFM surface. What is the physical mechanism on the improving electrochemical performances?
  3. In Fig. 3, why the colors of these NFM materials are differnt? Generally, the color becomes more black with the increasing C content.

Author Response

Response to Reviewer’s Comments

Manuscript ID: nanomaterials-1604937

TITLE: Solution Combustion Synthesis of NaFe0.5Mn0.5O2/C Composite as a Cathode for Improved Sodium-ion Storage Applications

Nanomaterials

     We thank the reviewer for his/her thorough review and highly welcome the comments and suggestions, which significantly contributed to improving the quality of the manuscript. The following responses (the modification parts are highlighted in blue colour in the revised manuscript) have been prepared to address the reviewer’s comments.

Reviewer 1

Comments and Suggestions for Authors

In this paper, the authors prepared composite layered NFM with carbon for the cathode electrode of SIBs through a facile solution combustion method followed by calcination process. This material presents an excellent electrochemical performance. This work is written well and organized reasonably. I wonder to recommend it for publication in Nanomaterials after some revisions.

Comment 1: This work provided the XPS data in fig. 5. It is better if the authors could add some discussion on the interface adhesion properties between C and NFM. Please see the references: Extreme Mechanics Letters 9 (2016) 226-236; Journal of Power Sources 290 (2015) 114-122.

Author’s response: we would like to thank the reviewer for his valuable comment. We have gone through the above references given by the reviewer and the calculations on the interface adhesion properties between C and NFM using the parameters from the XPS results. As reviewer knows well the method is time consuming one. Due to the lack of time we couldn’t find the interfacial adhesion using XPS analysis. We express our sincere apologies to the reviewer. We add one research work as a Reference in the revised manuscript.

Comment 2: The carbon particles only attach to the NFM surface. What is the physical mechanism on the improving electrochemical performances?

Author’s response: We would like to bring the reviewer’s kind attention to the following points: In general, as evidenced from the previous reports, composite preparation is considered a potential strategy to increase the structural stability and the conductive media of the active material. During the insertion/reinsertion of sodium ions, the degradation of the active material that reacts with electrolyte ions can be reduced by making the composite with heteroatoms. Carbon composite intensely improves interfacial stability and electric conductivity. The direct contact leads to unwanted side reactions from the electrolyte to the active material, thereby causing degradation of the electrode and subsequently, capacity fading. As seen in FE-SEM, the carbon particles are attached to the NFM surface, which restricts the direct contact between the active material and the electrolyte. In this study, the rate performance of the coating material has shown an improvement when compared with the pristine material. Therefore, the improved rate performances are observed due to the facile movement of ions and charges as evidenced from the electrochemical studies. Carbon composite intensely improves interfacial stability and electric conductivity.

Comment 3: In Fig. 3, why the colors of these NFM materials are different? Generally, the color becomes more black with the increasing C content.

Author’s response: We would like to bring the reviewer’s kind attention, as the reviewer well knows about SEM characterization. The color of the SEM images for all materials will be black in color. To easy understanding for the readers, the authors gave different colors for each sample to distinguish the as-prepared NFM, NFM/C-1, NFM/C-2, and NFM/C-3 materials. Many published articles were used this coloring for the better understanding to their reader. There are no other reasons for the color differences in NFM materials.

Reviewer 2 Report

Dear Authors:

In my viewpoint, the manuscript “Solution Combustion Synthesis of NaFe0.5Mn0.5O2/C Composite as a Cathode for Improved Sodium-ion Storage Applications” is interesting but cannot be accepted to publication. In this sense, if the ranking the manuscript went “major corrections” level will lead only delaying the decision-making process.  Then, after further revision the manuscript would be re-submitted receiving a new number. This ranking can be justified by comments below. As a function of a fast addressing of the comments, I divide comments in two parts, general comments and specific comments, in according with major topics of manuscript.

General comments:

As a whole, the English level needs be improved. In some parts, it is notable a bad (awkward) grammatical constructions. This mean that words appears in English but AS IST does not make sense. See as an example, in the Conclusion item:

“O3-type layered NFM cathode materials have been synthesized through a facile solution combustion technique for sodium-ion storage”. Here, “…sodium-ion storage” is vague and it is not a synonymous, since that the word storage is poorly related to a “battery”.

At moment, the diagramming is unsatisfactory . First, As ist the manuscript layout is very unusual. Second, the classical apprach should be engineered along manuscript, Figures should appears close to its presentation and discussion. Third, all "nano" aspects should be put in clear way along of manuscript. As a matter of fact, the prefix nano is used in a very very small number of times; but the Journal is at about NANOmaterials.

Specific comments:

Title item:

At moment, the title is trunked. I suggest a direct and most simple approach. As an example:

Preparation of Nanocomposite of type Carbon/Oxide in two steps, a Potential Material for Cathode Application in Sodium-ion type Batteries.

Abstract item:

Here, is clear the erroneous use of acronyms. First, it is not necessary use an acronym if the set of words be used only written one time, in a major item of text. The Sodium-Ion Batteries does not require the acronym (SIB).

Second, the acronym NFM is orphan, since does not is accompanied of its designator. In another words, NFM is not the acronym of “O3-type layered”.

Introduction item:

Authors are invited to carry out a deep revision of the English. This revision should be critical reaching the proper choose of verbs. As an example, the verb to introduce isn’t used in the context of this manuscript.

Another question involves the using of the plural that should be approached with careful. Batteries, a priori, make mention to a set of equal batteries. Batteries that are different in project but have operation based in a same material are further mentioned as Rechargeable batteries “type”.

At moment, there is evidence of language vicious. See, it is not necessary use “The authors”. In tens of manuscript I don’t see nothing similar this. In a general way, this paragraph and subsequent one exhibits a classical structure:

In this study, O3-type layered NaFe0.5Mn0.5O2 (NFM) crystalline and nanostructured oxide was synthesized via combustion method followed by calcination. The as-synthesized material was applied as a cathode for the SIB half-cell. The material exhibits higher specific charge capacity in initial cycles at a lower current rate.

However, it suffers poor rate capability and drastic capacity fading when cycled at higher current rates, which limits its potential applications. To overcome these limitations, the authors introduced carbon into the host NFM material to form a composite, which was then used as the cathode in sodium-ion energy storage device. Preparation of C/NFM active material using solution combustion method will be also newly discussed.

Well, there are several problems in a set of concepts used. The first estrange term is “specific capacity” that is applied in a broad sense to the thermodynamic area; in the electrochemistry seems more proper to make use “specific charge capacity”. The other unconventional term is current rate or lower current rate or yet higher current.

I suggest cutting some adverbs or great part of them. This procedure should allow a text close to an academic text. See, some adverbs identified, as follow: commercially and significantly, as well as intensively. Also, I think that it is possible see more correct use of  “as well as”.

The following phrase should be deleted: “To the best of the authors’ knowledge, this is the first time that the physical, chemical, and electrochemical properties of as-synthesized NFM/carbon composite material have been investigated and reported.” I believe that such type of phrase should be inserted in “Submission Letter” of a manuscript.

Results & Discussion item:

The characterization technique called X-ray diffraction doesn’t exhibits “peaks”; otherwise have a set of hkl planes called diffraction lines. Since, the full set of diffraction lines is defined the average crystallite size and lattice parameters can be derived and reported.

Conclusion Item:

The following phrase: “Because the NFM material has poor cyclic and rate performance, carbon composites were prepared by the solid-state method followed by calcination.” does not is a conclusion. As a matter of fact, this phrase seems one of justificative of the investigation. Then, in part, this phrase is further used in the Introduction item.

I suggest that this item should be rewritten. As a matter of fact, there is only a part that is similar to a conclusion that is “Therefore, the carbon composite helps to improve the stability of NFM cathode materials at higher current rates due to the increase of the electrical conductivity as well as the Na+ mobility for sodium-ion storage and the NFM/C composite is a promising candidate for high performance SIBs in the future.”

However, again appears a set words, terms and awkward grammatical constructions, as follow stability (?), as well as (?) and sodium-ion storage (?).

Also, it is possible to identify another three problems. The first problem is use the verb “to help” that seems inadequate. The second problem is the use of an acronym far from letters definition. The third one is assigned again to English, see in the phrase highlight that “the NFM/C composite is a promising candidate for high performance SIBs”, is not necessary to add “…in the future”.

Author Response

Response to Reviewer’s Comments

Manuscript ID: nanomaterials-1604937

TITLE: Solution Combustion Synthesis of NaFe0.5Mn0.5O2/C Composite as a Cathode for Improved Sodium-ion Storage Applications

Nanomaterials

     We thank the reviewer for his/her thorough review and highly welcome the comments and suggestions, which significantly contributed to improving the quality of the manuscript. The following responses (the modification parts are highlighted in blue colour revised manuscript) have been prepared to address the reviewer’s comments.

Reviewer 2

Comments and Suggestions for Authors

Dear Authors:

In my viewpoint, the manuscript “Solution Combustion Synthesis of NaFe0.5Mn0.5O2/C Composite as a Cathode for Improved Sodium-ion Storage Applications” is interesting but cannot be accepted to publication. In this sense, if the ranking the manuscript went “major corrections” level will lead only delaying the decision-making process.  Then, after further revision the manuscript would be re-submitted receiving a new number. This ranking can be justified by comments below. As a function of a fast addressing of the comments, I divide comments in two parts, general comments and specific comments, in according with major topics of manuscript.

General comments:

 Comment 1: As a whole, the English level needs be improved. In some parts, it is notable a bad (awkward) grammatical constructions. This mean that words appears in English but AS IST does not make sense. See as an example, in the Conclusion item:

“O3-type layered NFM cathode materials have been synthesized through a facile solution combustion technique for sodium-ion storage”. Here, “…sodium-ion storage” is vague and it is not a synonymous, since that the word storage is poorly related to a “battery”.

Author’s response: we would like to bring the reviewer’s kind attention, that the battery is a device that stores chemical energy and converts it to electrical energy. In the battery, we are storing the chemical energy and that will convert to electrical energy in the required time. As a matter of fact, battery is a storage device. So, in the above-mentioned place, the word storage is the most proper word in the electrochemistry field. Most of the published articles commonly used this word for battery applications.  Please kindly consider this viewpoint about the term.

Comment 2: At moment, the diagramming is unsatisfactory. First, As ist the manuscript layout is very unusual. Second, the classical apprach should be engineered along manuscript, Figures should appears close to its presentation and discussion. Third, all "nano" aspects should be put in clear way along of manuscript. As a matter of fact, the prefix nano is used in a very very small number of times; but the Journal is at about NANOmaterials.

Author’s response: We thank the reviewer for this valuable suggestion. We have prepared the manuscript according to the nanomaterials journal template advised by the journal.

As per the reviewer’s suggestion, the prefix “nano” has been used in the revised manuscript in the appropriate places as many as possible.

Specific comments:

 Title item:

Comment 3: At moment, the title is trunked. I suggest a direct and most simple approach. As an example:

Preparation of Nanocomposite of type Carbon/Oxide in two steps, a Potential Material for Cathode Application in Sodium-ion type Batteries.

Author’s response: With your thankful suggestions We authors have deeply thought to reflect the reviewer’s viewpoint and finally would like to give a new title as ‘Enhanced NaFe0.5Mn0.5O2/C Nanocomposite as a Cathode for Sodium-ion Batteries’.

Abstract item:

Comment 4:  Here, is clear the erroneous use of acronyms. First, it is not necessary use an acronym if the set of words be used only written one time, in a major item of text. The Sodium-Ion Batteries does not require the acronym (SIB).

Second, the acronym NFM is orphan, since does not is accompanied of its designator. In another words, NFM is not the acronym of “O3-type layered”.

Author’s response: we would like to bring the reviewer’s kind attention, that the word “sodium-ion batteries” is used many times in the manuscript. So, we have given the acronym for that word as SIBs which is widely used by the researchers.

The acronym NFM has been corrected in the revised manuscript.

Introduction item:

Comment 5: Authors are invited to carry out a deep revision of the English. This revision should be critical reaching the proper choose of verbs. As an example, the verb to introduce isn’t used in the context of this manuscript.

Author’s response: We thank the reviewer for this suggestion, we have checked carefully and corrected some erroneous expressions in the revised manuscript. And the manuscript had been edited by Editage (www.editage.co.kr) for an English language editing company.

Comment 6: Another question involves the using of the plural that should be approached with careful. Batteries, a priori, make mention to a set of equal batteries. Batteries that are different in project but have operation based in a same material are further mentioned as Rechargeable batteries “type”.

 Author’s response: We agree with the reviewer’s point. In that sentence, in order to mention all the different types of rechargeable batteries among the other energy storage devices we have commonly used that plural word as batteries.

Comment 7: At moment, there is evidence of language vicious. See, it is not necessary use “The authors”. In tens of manuscript I don’t see nothing similar this. In a general way, this paragraph and subsequent one exhibits a classical structure:

Author’s response: We thank the reviewer for this comment, and the term authors has removed in the revised manuscript.

Comment 8: In this study, O3-type layered NaFe0.5Mn0.5O2 (NFM) crystalline and nanostructured oxide was synthesized via combustion method followed by calcination. The as-synthesized material was applied as a cathode for the SIB half-cell. The material exhibits higher specific charge capacity in initial cycles at a lower current rate.

However, it suffers poor rate capability and drastic capacity fading when cycled at higher current rates, which limits its potential applications. To overcome these limitations, the authors introduced carbon into the host NFM material to form a composite, which was then used as the cathode in sodium-ion energy storage device. Preparation of C/NFM active material using solution combustion method will be also newly discussed.

 Well, there are several problems in a set of concepts used. The first estrange term is “specific capacity” that is applied in a broad sense to the thermodynamic area; in the electrochemistry seems more proper to make use “specific charge capacity”. The other unconventional term is current rate or lower current rate or yet higher current.

Author’s response: We thank the reviewer for suggesting this correction, the term has been modified as specific charge/discharge capacity in the revised manuscript.

Comment 9: I suggest cutting some adverbs or great part of them. This procedure should allow a text close to an academic text. See, some adverbs identified, as follow: commercially and significantly, as well as intensively. Also, I think that it is possible see more correct use of “as well as”.

 Author’s response: we thank reviewer for suggesting this correction. As per the reviewer’s suggestion, we have changed some adverbs in the revised manuscript.

Comment 10: The following phrase should be deleted: “To the best of the authors’ knowledge, this is the first time that the physical, chemical, and electrochemical properties of as-synthesized NFM/carbon composite material have been investigated and reported.” I believe that such type of phrase should be inserted in “Submission Letter” of a manuscript.

 Author’s response: We would like to bring the reviewer’s kind attention. The above statement was added in the manuscript for the reason to get the reader’s attention towards the work. As in the previously published articles, it is the common statement to prove the novelty of the work and new attempt that will get more attention from the readers.  

Results & Discussion item:

Comment 11: The characterization technique called X-ray diffraction doesn’t exhibits “peaks”; otherwise have a set of hkl planes called diffraction lines. Since, the full set of diffraction lines is defined the average crystallite size and lattice parameters can be derived and reported.

Author’s response: We thank the reviewer for this valuable comment. As per the reviewer’s suggestion, we have derived the average crystalline size through the “Scherrer equation” by using the lattice parameters. The average crystalline size of the NFM nanocomposite is 42.69 nm. This value is added in the revised version of the manuscript. 

Conclusion Item:

Comment 12: The following phrase: “Because the NFM material has poor cyclic and rate performance, carbon composites were prepared by the solid-state method followed by calcination.” does not is a conclusion. As a matter of fact, this phrase seems one of justificative of the investigation. Then, in part, this phrase is further used in the Introduction item.

I suggest that this item should be rewritten. As a matter of fact, there is only a part that is similar to a conclusion that is “Therefore, the carbon composite helps to improve the stability of NFM cathode materials at higher current rates due to the increase of the electrical conductivity as well as the Na+ mobility for sodium-ion storage and the NFM/C composite is a promising candidate for high performance SIBs in the future.”

Author’s response: As per the reviewer’s suggestion, the conclusion has been modified a bit in the revised version of the manuscript.

Comment 13: However, again appears a set words, terms and awkward grammatical constructions, as follow stability (?), as well as (?) and sodium-ion storage (?).

Author’s response: As per the reviewer’s suggestion, the above-mentioned comments were considered and modified in the revised version of the manuscript.

Comment 14: Also, it is possible to identify another three problems. The first problem is use the verb “to help” that seems inadequate. The second problem is the use of an acronym far from letters definition. The third one is assigned again to English, see in the phrase highlight that “the NFM/C composite is a promising candidate for high performance SIBs”, is not necessary to add “…in the future”.

Author’s response: As per the reviewer’s suggestion, the above mentioned-comments were considered and modified in the revised version of the manuscript.

Reviewer 3 Report

Lithium-ion batteries are one of the emerging energy storage devices that recently attracted wide attention, this is due to their potential benefits in the market. The current generation of Sodium-ion batteries is also playing a great role and is considered a viable alternative to ubiquitous lithium-ion batteries, owing to their significant cost advantages stemming from the high natural abundance and broad distribution of sodium resources. Authors have reported O3-type layered NaFe0.5Mn0.5O2 (NFM) material through a facile solution combustion technique followed by calcination.

The manuscript is well presented with supportive synthesis, physical and battery electrochemical validations exhibiting higher discharge capacities; it can be considered after some revision. Some parts of the manuscript can be improved.

Following are my specific comments:

  • In the abstract, line 13, what is NFM – define?
  • In the abstract, please also include the charge/discharge capacities
  • In the introduction, line 30, what is SIB?
  • In the introduction, line 36, the word scientists can be replaced by “Researchers in the energy storage community”
  • In the introduction, lines 40 -43, apart from layered transition metal oxides, binary metal oxides are also widely used in SIBs as reported by Manickam Minakshi group.
  • The different ways of the stacking of the oxygen arrangement in P2- and O3-type structures need to be detailed a bit more.
  • Will the diffusion of Na ions from one octahedral site to another would take place through the edge-sharing tetrahedral site between them? Because the occupation of Na+ in the tetrahedral sites is unfavorable due to the size discrepancy. Does the purpose of Fe is to improve these sluggish kinetics?
  • How do the sodium storage properties of current work in NFM compare with O3-NaNi0.5Mn0.5O2 (NNM)?
  • What is the role of oxidizer/fuel in combustion synthesis?
  • Why the cut-off voltage for the charge has been as high as 4.3 V? Is the electrolyte safe?
  • The reported NFM materials need to be benchmarked against high-performance sodium devices such as 10.1039/C8NR03824D; 10.1039/C8NA00156A.
  • Does the formation of a composite with carbon arises from the chosen synthesis technique and the presence of fuel?

Author Response

Response to Reviewer’s Comments

Manuscript ID: nanomaterials-1604937

TITLE: Solution Combustion Synthesis of NaFe0.5Mn0.5O2/C Composite as a Cathode for Improved Sodium-ion Storage Applications

Nanomaterials

     We thank the reviewer for his/her thorough review and highly welcome the comments and suggestions, which significantly contributed to improving the quality of the manuscript. The following responses (the modifications are highlighted in blue colour in the revised manuscript) have been prepared to address the reviewer’s comments.

Reviewer 3:

Following are my specific comments:

Comment 1: In the abstract, line 13, what is NFM – define?

 Author’s response: We thank the reviewer for suggesting this correction. As per the reviewer’s suggestion, we have defined NFM at line 13 of Abstract in the revised manuscript.

Comment 2: In the abstract, please also include the charge/discharge capacities

 Author’s response: As per the reviewer’s suggestion, we have included the specific charge/discharge capacity for NFM/C-2 material in the revised version of the manuscript.

Comment 3: In the introduction, line 30, what is SIB?

Author’s response: We would like to thank the reviewer for his comment. The abbreviation of SIB has already been defined in the Abstract section.

Comment 4: In the introduction, line 36, the word scientists can be replaced by “Researchers in the energy storage community”

Author’s response: As per the reviewer’s suggestion, the word scientists has been replaced by “Researchers in the energy storage community” in the revised manuscript.

Comment 5: In the introduction, lines 40 -43, apart from layered transition metal oxides, binary metal oxides are also widely used in SIBs as reported by Manickam Minakshi group.

Author’s response: We would like to thank the reviewer for this comment, and we agree with the reviewer’s comment. As mentioned by the reviewer, binary metal oxides are also widely used in sodium-ion batteries but as an anode. As a cathode a very small amount of binary metal oxides study was carried out and produced a moderate specific capacity. In SIBs, the layered sodium transition metal oxides are widely used as a cathode. That’s the reason we haven’t mentioned binary metal oxides in lines 40-43.

Comment 6: The different ways of the stacking of the oxygen arrangement in P2- and O3-type structures need to be detailed a bit more.

Author’s response: We would like to thank the reviewer for this comment. As per the reviewer’s suggestion, the stacking of the oxygen arrangement in P2 and O3-type structures were included in the revised manuscript.   

Comment 7: Will the diffusion of Na ions from one octahedral site to another would take place through the edge-sharing tetrahedral site between them? Because the occupation of Na+ in the tetrahedral sites is unfavorable due to the size discrepancy. Does the purpose of Fe is to improve these sluggish kinetics?

Author’s response: We would like to thank the reviewer for this comment. Despite of inferior energy densities of SIBs, inexpensive and Ni/Co-free Na-ion cathode materials can enable commercial viability of SIBs as LIB technology is at the risk of Ni and Co supply. The use of earth-abundant transition metals like Fe and Mn as redox reaction metals can further reduce the cost of SIBs. Layered NaMnO2 provides significant energy density, and Fe is electrochemically active in the Na-ion cathode. O3-FeMn-based materials with the advantages of nontoxic and storage abundance in the earth’s crust are the key motivation in the exploration of NFM cathodes for SIBs in this work. The purpose of Fe in this work is to achieve synergetic electrochemical advances. The partial Fe doping in NaMnO2 will provide higher initial capacities [1-3]. We planned this work to produce a potential cathode for upcoming SIBs by considering the above points.

References:

  1. Billaud, Juliette, et al. "β-NaMnO2: a high-performance cathode for sodium-ion batteries." Journal of the American Chemical Society 136.49 (2014): 17243-17248.
  2. Li, Yanyang, et al. "From α-NaMnO 2 to crystal water containing Na-birnessite: enhanced cycling stability for sodium-ion batteries." CrystEngComm 18.17 (2016): 3136-3141.
  3. Clement, Raphaele J., et al. "Insights into the nature and evolution upon electrochemical cycling of planar defects in the β-NaMnO2 Na-ion battery cathode: An NMR and first-principles density functional theory approach." Chemistry of Materials 28.22 (2016): 8228-8239.

Comment 8: How do the sodium storage properties of current work in NFM compare with O3-NaNi0.5Mn0.5O2 (NNM)?

Author’s response: We would like to thank the reviewer for the comment. O3-type layered transition metal oxides have been well-known as the promising cathode material for sodium-ion batteries due to their high theoretical capacity. Still, they seriously suffer from inferior cycling stability and poor rate capability due to complicated phase transitions during the cycling. To overcome these limitations, the authors introduced carbon into the NFM material to form a nanocomposite, which is then used as the cathode in sodium-ion batteries. To the best of the authors’ knowledge, this is the first time that the physical, chemical, and electrochemical properties of as-synthesized NFM/carbon composite material have been investigated and reported. The NFM/carbon composite with 3 wt.% carbon precursor exhibited the best performance, with a specific discharge capacity of 188.20 mAh g-1 at 0.05 C and 145.27 mAh g-1 at 0.1 C. Whereas, NNM material reported by Q. Mao et al. delivered a specific discharge capacity of 133 mAh g-1 at 0.075 C [1]. This implies that NFM material exhibited better performance compared to NNM in terms of specific discharge capacity.

Reference:

  1. Mao, Qianjiang, et al. "O3-type NaNi0. 5Mn0. 5O2 hollow microbars with exposed {0 1 0} facets as high performance cathode materials for sodium-ion batteries." Chemical Engineering Journal 382 (2020): 122978.

Comment 9: What is the role of oxidizer/fuel in combustion synthesis?

Author’s response: We would like to bring the reviewer’s kind attention, the solution combustion synthesis is an efficient synthesis technique in terms of cost-effectiveness, easiness, and the quality of the product. It is an exothermic and self-sustained redox reaction between a fuel and oxidizers in the presence of metal cations. Generally, oxidizers are metal precursors such as metal nitrates sulfates, carbonates, etc., and fuels are organic materials such as citric acid, urea, glycine, etc. [1]. The fuel has the central role in the optimization of the product materials properties, and it has the triple function of reducer, complexing agent, and microstructural template. The nature of the fuel and fuel to oxidizer ratio play important roles in combustion synthesis since they can influence the morphology, phase, and particulate properties of the final combusted product [2,3].

References:

  1. Sherikar, Baburao N., Balaram Sahoo, and Arun M. Umarji. "Effect of fuel and fuel to oxidizer ratio in solution combustion synthesis of nanoceramic powders: MgO, CaO and ZnO." Solid State Sciences 109 (2020): 106426.
  2. Deganello, Francesca, and Avesh Kumar Tyagi. "Solution combustion synthesis, energy and environment: Best parameters for better materials." Progress in Crystal Growth and Characterization of Materials 64.2 (2018): 23-61.
  3. Challagulla, Swapna, and Sounak Roy. "The role of fuel to oxidizer ratio in solution combustion synthesis of TiO2 and its influence on photocatalysis." Journal of Materials Research 32.14 (2017): 2764-2772.

Comment 10: Why the cut-off voltage for the charge has been as high as 4.3 V? Is the electrolyte safe?

Author’s response: We would like to bring the reviewer’s kind attention. Generally, layered-type, high-energy density cathode materials for sodium-ion batteries are safe to use in a wide range of potential window (1.5-4.5 V). Also, it is previously reported that there are several organic electrolytes that are stable in the potential window of 0-5.0 V. We have used 1 M NaClO4 in a mixture of ethylene carbonate and diethylene carbonate (1:1 volume ratio) with 5 wt.% fluoroethylene carbonate as an electrolyte for this present work, and as per the previous reports it is stable in the potential window of 1.5₋4.3 V. Additionally, here we are providing some references to support our point [1-3].

References:

  1. Ponrouch, Alexandre, et al. "In search of an optimized electrolyte for Na-ion batteries." Energy & Environmental Science 5.9 (2012): 8572-8583.
  2. Lin, Zeheng, et al. "Recent research progresses in ether‐and ester‐based electrolytes for sodium‐ion batteries." InfoMat 1.3 (2019): 376-389.
  3. Ponrouch, A., et al. "Non-aqueous electrolytes for sodium-ion batteries." Journal of Materials Chemistry A 3.1 (2015): 22-42.

Comment 11: The reported NFM materials need to be benchmarked against high-performance sodium devices such as 10.1039/C8NR03824D; 10.1039/C8NA00156A.

Author’s response: We would like to thank the reviewer for the comment. We have studied both of the above-mentioned articles. In the first article, MgMoO4 was synthesized by the solution combustion method and used an anode for sodium-ion batteries. In the second article, CaMoO4 was fabricated by the solution combustion method and used an anode for sodium-ion batteries. In our manuscript, we have synthesized O3-type layered NaFe0.5Mn0.5O2/carbon (NFM/C) composite cathode material through a solution combustion technique for sodium-ion batteries.  The only similarity between the articles mentioned by the reviewer and the present work is the use of the same synthesis route. Our work is focused on the layered type cathode materials for sodium-ion batteries, which are considered as a potential candidate for future commercial sodium-ion batteries. Therefore, we believe that there is no relevance in comparing the current manuscript and the above-mentioned literature. However, we have cited the articles in the introduction part in the revised manuscript [26, 27].                                                                                                                           

Comment 12: Does the formation of a composite with carbon arises from the chosen synthesis technique and the presence of fuel?

Author’s response: We would like to appreciate the reviewer’s comment. The formation of NaFe0.5Mn0.5O2/carbon (NFM/C) composite is not because of the chosen synthesis technique and the presence of fuel. We have chosen the solution combustion synthesis method because of its characteristics such as cost-effectiveness, facile process, and the quality of the product. For solution combustion synthesis fuel acts as a reducing agent, complexing agent as well as the microstructural template. We developed an O3-type layered NaFe0.5Mn0.5O2 (NFM) material through a facile solution combustion technique followed by calcination. The carbon content from the use of fuel during the synthesis will be eliminated during the high-temperature calcination process in the air atmosphere. To overcome the inherent limitations associated with NFM material, we have introduced 2-methylamidazole as a carbon source to form an NFM/C composite.

Round 2

Reviewer 2 Report

Dear Authors:

In my viewpoint, the manuscript “Solution Combustion Synthesis of NaFe0.5Mn0.5O2/C Composite as a Cathode for Improved Sodium-ion Storage Applications” is interesting but cannot be accepted to publication. In this sense, if the ranking the manuscript went “major corrections” level will lead only delaying the decision-making process.  Then, after further revision the manuscript would be re-submitted receiving a new number. This ranking can be justified by comments below. As a function of a fast addressing of the comments, I divide comments in two parts, general comments and specific comments, in according with major topics of manuscript.

General comments:

As a whole, the English level needs be improved. In some parts, it is notable a bad (awkward) grammatical constructions. This mean that words appears in English but AS IST does not make sense. See as an example, in the Conclusion item:

“O3-type layered NFM cathode materials have been synthesized through a facile solution combustion technique for sodium-ion storage”. Here, “…sodium-ion storage” is vague and it is not a synonymous, since that the word storage is poorly related to a “battery”.

Specific comments:

Title item:

At moment, the title is trunked. I suggest a direct and most simple approach. As an example:

Preparation of Nanocomposite of type Carbon/Oxide in two steps, a Potential Material for Cathode Application in Sodium-ion type Batteries.

Abstract item:

Here, is clear the erroneous use of acronyms. First, it is not necessary use an acronym if the set of words be used only written one time, in a major item of text. The Sodium-Ion Batteries does not require the acronym (SIB).

Second, the acronym NFM is orphan, since does not is accompanied of its designator. In another words, NFM is not the acronym of “O3-type layered”.

Introduction item:

Authors are invited to carry out a deep revision of the English. This revision should be critical reaching the proper choose of verbs. As an example, the verb to introduce isn’t used in the context of this manuscript.

Another question involves the using of the plural that should be approached with careful. Batteries, a priori, make mention to a set of equal batteries. Batteries that are different in project but have operation based in a same material are further mentioned as Rechargeable batteries “type”.

At moment, there is evidence of language vicious. See, it is not necessary use “The authors”. In tens of manuscript I don’t see nothing similar this. In a general way, this paragraph and subsequent one exhibits a classical structure:

In this study, O3-type layered NaFe0.5Mn0.5O2 (NFM) crystalline and nanostructured oxide was synthesized via combustion method followed by calcination. The as-synthesized material was applied as a cathode for the SIB half-cell. The material exhibits higher specific charge capacity in initial cycles at a lower current rate.

However, it suffers poor rate capability and drastic capacity fading when cycled at higher current rates, which limits its potential applications. To overcome these limitations, the authors introduced carbon into the host NFM material to form a composite, which was then used as the cathode in sodium-ion energy storage device. Preparation of C/NFM active material using solution combustion method will be also newly discussed.

Well, there are several problems in a set of concepts used. The first estrange term is “specific capacity” that is applied in a broad sense to the thermodynamic area; in the electrochemistry seems more proper to make use “specific charge capacity”. The other unconventional term is current rate or lower current rate or yet higher current.

I suggest cutting some adverbs or great part of them. This procedure should allow a text close to an academic text. See, some adverbs identified, as follow: commercially and significantly, as well as intensively. Also, I think that it is possible see more correct use of  “as well as”.

The following phrase should be deleted: “To the best of the authors’ knowledge, this is the first time that the physical, chemical, and electrochemical properties of as-synthesized NFM/carbon composite material have been investigated and reported.” I believe that such type of phrase should be inserted in “Submission Letter” of a manuscript.

Results & Discussion item:

The characterization technique called X-ray diffraction doesn’t exhibits “peaks”; otherwise have a set of hkl planes called diffraction lines. Since, the full set of diffraction lines is defined the average crystallite size and lattice parameters can be derived and reported.

Conclusion Item:

The following phrase: “Because the NFM material has poor cyclic and rate performance, carbon composites were prepared by the solid-state method followed by calcination.” does not is a conclusion. As a matter of fact, this phrase seems one of justificative of the investigation. Then, in part, this phrase is further used in the Introduction item.

I suggest that this item should be rewritten. As a matter of fact, there is only a part that is similar to a conclusion that is “Therefore, the carbon composite helps to improve the stability of NFM cathode materials at higher current rates due to the increase of the electrical conductivity as well as the Na+ mobility for sodium-ion storage and the NFM/C composite is a promising candidate for high performance SIBs in the future.”

However, again appears a set words, terms and awkward grammatical constructions, as follow stability (?), as well as (?) and sodium-ion storage (?).

Also, it is possible to identify another three problems. The first problem is use the verb “to help” that seems inadequate. The second problem is the use of an acronym far from letters definition. The third one is assigned again to English, see in the phrase highlight that “the NFM/C composite is a promising candidate for high performance SIBs”, is not necessary to add “…in the future”.

Author Response

Letter of Rebuttal

Dear Editor, Katarina Nesovic

Manuscript ID: nanomaterials-1604937

Title: Solution Combustion Synthesis of NaFe0.5Mn0.5O2/C Composite as a Cathode for Improved Sodium-ion Storage Applications

Authors: Nanthagopal Murugan, Chang Won Ho, Nitheesha Shaji, Gyu Sang Sim, Varun Karthik Murugesan, Hong Ki Kim, Chang Woo Lee *

We received a request for the second revision of our manuscript from Reviewer 2. Reviewer 2 had raised 14 Comments in the first revision. However, Reviewer 2 gave us 13 Comments in this 2nd revision, and those 13 Comments are exactly the same, copying and pasting the previous Comments suggested to us, as the Comments in the 1st revision process. 

We agree and respect the Editor’s decision and reviewers’ comments. However, we have responded according to the comments raised by the Reviewers in the first round of revision. The Reviewer said that the response has not been provided sufficiently by the authors in the first revision. But, he didn’t mention the response we submitted from our side in the first revision for the same comments. This is the first time we are getting the same comments without saying anything about the response and revised manuscript. If Reviewer 2 is not satisfied or the revision is not sufficient, he might have mentioned what is wrong in the response and revised manuscript.

All authors have considered the reviewer's comments very carefully and responded to the Reviewers Comments scientifically and reasonably in the first revision. The authors decided to appeal the reconsideration and also send a rebuttal in order to prevent recurrence in the future.

For example, we’d like to rebut the Reviewers’ 2 comments as follows. He said that the manuscript needs English correction. We agreed to the Reviewer in some places needs to change and the terms suggested by the reviewer had been revised in the revised manuscript according to the Reviewer’s suggestions even though the authors already edited the manuscript with the help of Editage (www.editage.co.kr) English language editing company for improving the quality of the manuscript. And we are submitting the Editing Certificate to prove that the manuscript had been edited by the global English editing company.     

The Reviewer commented that the word “storage” is not proper to use in the battery. As a matter of fact, the battery is a storage device. So, the word storage is the most proper word in the electrochemistry field. Most of the published articles commonly used this word for battery applications.

Another suggestion from the Reviewer is to change the title of the manuscript. We agreed and modified the previous title according to the research article. The authors have deeply thought to reflect the reviewer’s viewpoint and finally would like to give a new title as ‘Enhanced NaFe0.5Mn0.5O2/C Nanocomposite as a Cathode for Sodium-ion Batteries’

Therefore, we ask for your understanding that at this stage, we are submitting the Letter of Rebuttal.

Thank you for your kind reconsideration.

Chang Woo LEE

Reviewer 3 Report

The authors have responded to my queries satisfactorily. The revised part of the manuscript reads well. To this reviewer's opinion, the version is suitable to publish.

Author Response

We thank the reviewer for accepting the manuscript.